# Machine Learning Techniques for Vertical Lidar-Based Detection, Characterization, and Classification of Aerosols and Clouds: A Comprehensive Survey

Simone Lolli 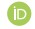

CNR-Institute of Methodologies for Environmental Analysis (IMAA), Contrada S. Loja snc, 85050 Tito Scalo, PZ, Italy; simone.lolli@cnr.it

**Abstract:** This survey presents an in-depth analysis of machine learning techniques applied to lidar observations for the detection of aerosol and cloud optical, geometrical, and microphysical properties. Lidar technology, with its ability to probe the atmosphere at very high spatial and temporal resolution and measure backscattered signals, has become an invaluable tool for studying these atmospheric components. However, the complexity and diversity of lidar technology requires advanced data processing and analysis methods, where machine learning has emerged as a powerful approach. This survey focuses on the application of various machine learning techniques, including supervised and unsupervised learning algorithms and deep learning models, to extract meaningful information from lidar observations. These techniques enable the detection, classification, and characterization of aerosols and clouds by leveraging the rich features contained in lidar signals. In this article, an overview of the different machine learning architectures and algorithms employed in the field is provided, highlighting their strengths, limitations, and potential applications. Additionally, this survey examines the impact of machine learning techniques on improving the accuracy, efficiency, and robustness of aerosol and cloud real-time detection from lidar observations. By synthesizing the existing literature and providing critical insights, this survey serves as a valuable resource for researchers, practitioners, and students interested in the application of machine learning techniques to lidar technology. It not only summarizes current state-of-the-art methods but also identifies emerging trends, open challenges, and future research directions, with the aim of fostering advancements in this rapidly evolving field.

**Keywords:** lidar; machine learning; aerosols; clouds; rain; boundary layer

## 1. Introduction

The light detection and ranging (lidar) technique has emerged as a critical tool for understanding the Earth's atmosphere. It enables retrieval with high temporal and spatial resolution of the vertically resolved optical, microphysical, and geometric properties of aerosols [1,2] and clouds [3,4], which play a key role in modulating the Earth's radiation budget [5,6], influencing climate change, and affecting air quality [7–10]. Figure 1 shows the main domains in which lidar technology plays a crucial role in atmospheric studies. Aerosols and clouds are dynamic entities that constantly change in response to various environmental and anthropogenic factors. Their intricate relationships with radiation, weather patterns, and climate processes [11] require the use of vertically resolved high-resolution atmospheric profiles. Moreover, capturing data at a high temporal resolution allows for detailed and timely observations of atmospheric changes. Such comprehensive insights are uniquely offered by technologies such as ground-based lidar. On the contrary, lidar instruments deployed on satellite, aircraft, and ground-based platforms are capable of providing near-continuous vertical profiles (with a much longer revisit time) of aerosols and clouds [12,13], allowing researchers to gain an unprecedented view of their

structure, density, and distribution. However, while lidar technology continues to evolve and improve, the complexity and sheer volume of data it generates present considerable challenges. The heterogeneity and dynamic nature of aerosols and clouds make the retrieval of their properties a complex task. Here, machine learning comes into play. Machine learning, a subset of artificial intelligence [14,15], can handle large datasets and unravel intricate patterns within them. It offers robust methodologies for automating the extraction of meaningful information from complex data. Machine learning algorithms can extract valuable insights from lidar observations, to characterize and detect cloud and aerosol layers. This aids in their parameterization for climate and weather forecasting models and advances our understanding of their impacts on climate and air quality. In the past two decades, the literature has witnessed numerous classic approaches, but in recent years, analysis through machine learning algorithms has presented a promising path toward a better and more comprehensive understanding of aerosol and cloud dynamics, as well as their broader environmental implications. In this survey, the author analyzes the state of the art of the principal machine learning and artificial intelligence (AI) techniques applied to lidar observations to obtain aerosol and cloud properties.

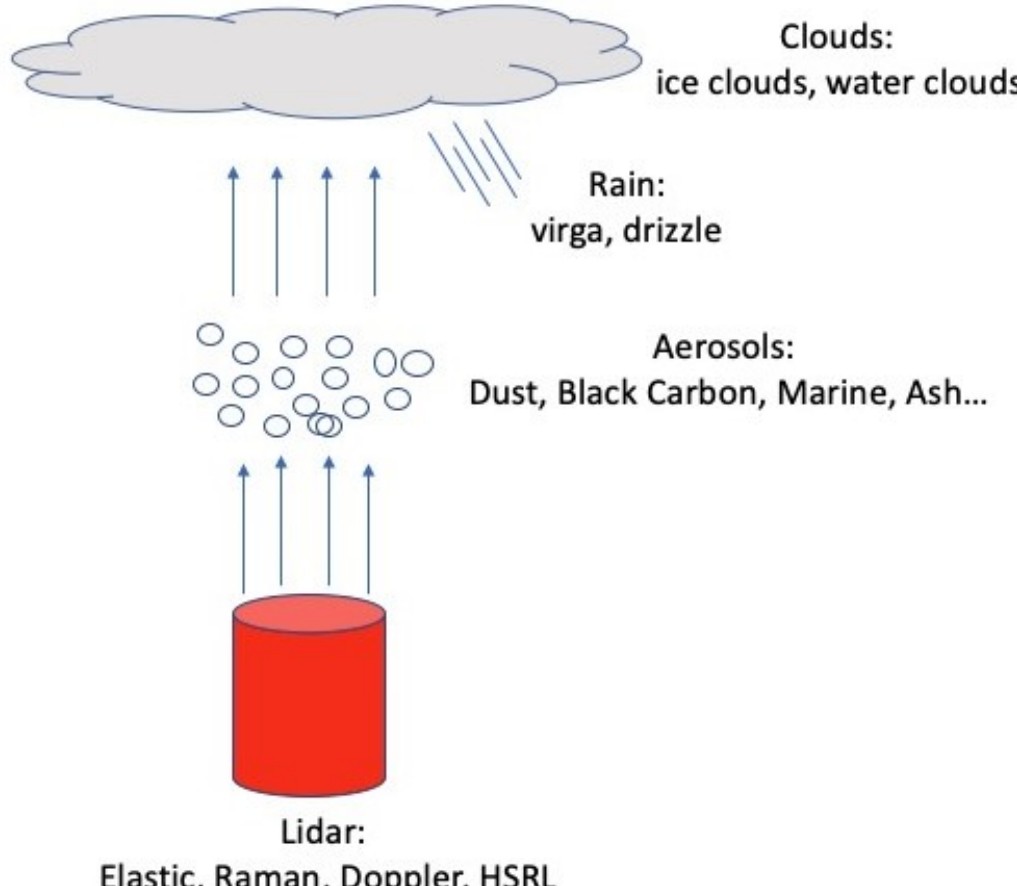

**Figure 1.** Different lidar techniques and relative detection/classification of aerosols, clouds, and precipitation problems where machine learning algorithms can play a key role.

This survey is primarily aimed at students and professionals seeking to deepen their understanding of the topic, providing a detailed exploration of the fundamental aspects of machine learning applied to lidar data analysis, and categorizing the approaches into three main classes. The first class revolves around the detection and classification of aerosols, clouds, and precipitation. The second class focuses on detecting mixing layer height (MLH) and planetary boundary layer height (PBLH) by incorporating machine learning and deep learning approaches borrowed from computer vision. Hybrid approaches that combine different philosophies have also been considered within this category. This survey aims

to change perspectives on quality assessment, by encouraging the application of newly developed approaches in real-world environments. In doing so, the aim is to outline new directions for future research in this domain.

In conclusion, this survey aims to provide a comprehensive overview of the machine learning technologies applied to atmospheric physics lidar observations. It caters to students and professionals interested in the topic, presenting the foundational aspects. This survey aims to stimulate further research in this field and promote a fresh perspective on these emerging methodologies.

## 2. Materials and Methods

### 2.1. Lidar Technology

Lidar techniques offer a diverse range of capabilities for studying aerosols and clouds. Elastic lidar (EL), or elastic backscatter lidar, is one of the most widely used methods for detecting and profiling aerosols. It measures backscattered light at the same wavelength as the emitted laser pulse, providing valuable information on aerosol layers and their vertical structure [6,16]. However, elastic lidar lacks the ability to derive specific aerosol properties, such as aerosol optical depth or extinction coefficient, without additional assumptions [17] or additional measurements. On the other hand, Raman lidar (RL) is a more sophisticated technique, named for the Raman scattering effect, that overcomes some of the limitations of elastic lidar. It measures the backscattered light at shifted wavelengths, which can be used to retrieve the aerosol extinction coefficient and other properties, without additional assumptions, such as the so-called lidar ratio [18,19]. RL can also provide information on water–vapor mixing ratio profiles, which is an essential parameter for understanding the hydrological cycle and cloud-formation processes. High-spectral-resolution laser (HSRL, [20]) is another technology that, thanks to a high-resolution spectral filter, provides direct and independent measurements of aerosol optical properties due to extinction and backscatter, allowing for more precise investigations of aerosol layers. The Doppler wind lidar (DWL) technique presents a different approach from the other techniques listed above. Instead of relying solely on the intensity or polarization of backscattered laser light, the DWL technique specifically analyzes the frequency shift between the transmitted and received light spectrum. This frequency shift, often known as the Doppler shift, is induced by the motion of particles (such as aerosols or molecules) in the atmosphere with which the light interacts. By processing this frequency difference through interference filters, DWL can accurately retrieve the wind speed along the line of sight. This method offers precise and direct measurement of wind profiles, making it an invaluable tool in meteorological observations and forecasting [21].

Each of these techniques presents unique capabilities and limitations, and the choice between them often depends on the specific requirements of the aerosol and cloud studies that are being undertaken and, of course, the available budget. In recent years, the advancement of lidar technology has increased the complexity of the field. As a result, extensive research on retrieval and denoising methods has become crucial to retrieve the properties of aerosol layers and clouds [22]. Lidar technology is traditionally used to evaluate and obtain the optical, microphysical, and geometric characteristics of aerosols and clouds. Furthermore, it is utilized to determine wind speed profiles, evaluate turbulence, and study processes within the boundary layer, which is the section of the troposphere directly in contact with the Earth's surface. Lidars are adept at differentiating between various types of aerosols and clouds, particularly through the lidar volume depolarization channel. Irregularly shaped entities such as dust particles and ice crystals scatter light in a highly depolarized fashion. On the contrary, other aerosols and clouds, such as water droplets or biomass burning, are more spherical and result in reduced depolarization of scattered light [16,23,24]. Notably, the aerosol classifications in CALIPSO do not consist of distinct categories but are rather continuous mixtures, making the achieved level of agreement commendable. Another pivotal application of lidar technology is the determination of the thickness of the boundary layer. As Stull elucidates [25], the boundary layer is the

atmospheric layer directly affected by the Earth's surface and that reacts to its forcings within roughly an hour. This layer exhibits turbulent flow, significantly influenced by friction against the Earth's surface. Processes such as turbulence, convection, and radiative transfer within the boundary layer facilitate the exchange of heat, moisture, and momentum between the Earth's surface and the free troposphere. Therefore, the thickness of the boundary layer is a sought-after parameter for various applications, such as dispersion models. Lidars are particularly apt for gauging the thickness of the planetary boundary layer or the mixing layer's thickness during the day when convection is evident. Typically, lidar uses the lowermost aerosol layer as an indicator to estimate the thickness of the boundary layer, as the aerosol concentration shows a pronounced shift in the transition to the free atmosphere. Traditional approaches determine the thickness of the boundary layer by gauging the gradient at this transition point [26] or employing more recent image processing methodologies [27].

For all the previously described applications, denoising methods are essential for lidar data processing, as they help mitigate the effects of noise and improve the accuracy of the retrieved information. Lidar measurements are susceptible to various sources of noise, including solar background light, detector noise, and atmospheric variability. Denoising techniques aim to enhance the signal-to-noise ratio (SNR), allowing for a clearer detection of aerosol layers and clouds. These methods often involve sophisticated filtering algorithms, statistical approaches, and signal processing techniques, to effectively remove unwanted noise, while preserving the underlying atmospheric characteristics [28]. In this context, machine learning algorithms, together with denoising techniques, are of paramount importance to improve the retrieval accuracy of the less sophisticated and cheaper lidar instruments that are often deployed in networks [29].

The raw signals acquired from the different lidar techniques, as defined in Table 1, frequently exhibit a considerable amount of noise. This noise is predominantly a challenge during daytime measurements, when the solar background significantly degrades the signal-to-noise ratio (SNR). To mitigate this issue and extract meaningful information from these signals, various denoising techniques are employed. The overarching objective of these algorithms is to improve the SNR. However, it is imperative that while the SNR is amplified, the intrinsic geometrical and physical properties of the aerosols and clouds remain undistorted. This careful balance ensures that the refined data retain their precision and relevance in atmospheric studies. Traditional denoising algorithms are summarized in Table 2.

**Table 1.** Summary of different lidar techniques for atmospheric studies.

| Lidar Technique | Physical Process | Remarks |
| --- | --- | --- |
| Elastic | Based on the elastic scattering of aerosols and clouds (Mie theory). Scattering is sensitive to particles about the same size as the wavelength of light and is responsible for the white appearance of clouds. | To solve the lidar equation, the lidar ratio is needed. |
| Raman | Based on Raman inelastic scattering, this technology is used to retrieve the properties of aerosols and clouds. It also provides temperature profiles and can distinguish between liquid water and ice clouds. | Aerosol and molecule properties can be recovered independently, without particular assumptions. Nevertheless, the system needs periodic calibrations. |
| HSRL | HSRL provides unambiguous separation of aerosol and cloud scattered signal, allowing for accurate aerosol optical depth measurements and better aerosol type discrimination. | The signals from molecules and aerosols are treated independently, with higher precision. However, this technology is quite expensive. |
| DWL | Based on Doppler shift, DWL analyzes the frequency shift between the transmitted and received light spectrum to retrieve the wind speed along the line of sight. | In addition to wind speed, DWL can also be used as an elastic lidar with the same limitations. |

**Table 2.** Traditional denoising methods for lidar techniques applied to atmospheric studies.

| Method | Description |
| --- | --- |
| Averaging | A simple method that involves averaging over several consecutive bins to enhance the signal-to-noise ratio. |
| Median Filtering | Replaces each bin in the profile with the median of neighboring bins to remove noise. |
| Wavelet Transform [30] | Decomposes the signal into wavelet coefficients at different scales for noise reduction at each scale. |
| Savitzky–Golay Filtering [31] | Uses polynomial regression to smooth out the data, while preserving the shape of the signal. |
| Gaussian Filtering [32] | Convolves the lidar signal with a Gaussian function for data smoothing. |
| High-pass and Low-pass Filtering [33] | Removes noise from specific frequency ranges; high-pass filters remove slow-varying drifts, and low-pass filters remove high-frequency noise. |
| Adaptive Filtering | Adjusts based on the signal and noise characteristics by dynamically changing its coefficients. |
| Statistical Methods | Removes those bins that deviate significantly from a statistical measure, assuming deviations from the mean are due to noise. |

*2.2. Main Machine Learning Algorithms*

The following sections describe the main machine learning techniques applied to lidar observations of aerosols and clouds.

2.2.1. Convolutional and Artificial Neural Networks

Convolutional neural networks (CNN) and artificial neural networks (ANN) have carved a niche for themselves in the realm of atmospheric science, exhibiting progress in both spatial and temporal data analysis [34]. CNNs are renowned for their ability to understand spatial hierarchies and associations. This makes them especially suitable for analyzing the satellite imagery obtained from remote sensing, and they have become a powerful tool for detection and classification tasks in various domains, ranging from computer vision to natural language processing [35]. Using convolutional layers, they are designed to capture spatial nuances such as cloud formations, storm patterns, and aerosol distributions [36]. For example, CNNs have shown efficacy in classifying cloud types from satellite images and predicting the intensity of tropical cyclones based on specific visual patterns [37]. However, ANNs, with their layered architecture, are instrumental in deciphering intricate relationships in large datasets. In atmospheric studies, ANNs are often used to forecast the weather, model climate, and identify specific meteorological events [38]. For example, training ANNs on historical weather data enables them to recognize patterns and relationships between disparate atmospheric variables, facilitating the prediction of future weather conditions. Furthermore, the adaptive nature of ANNs allows the identification and localization of atmospheric phenomena such as aerosol layers, cyclone centers, and storm fronts using datasets such as radar measurements or lidar scans [39]. It is essential to note that the efficacy of CNNs, ANNs, and machine learning techniques in general is intrinsically related to the quality and completeness of the training data and the architectural nuances tailored for specific tasks. This nexus of neural network technology and atmospheric science continues to foster advances that further our understanding of the dynamic atmosphere of the Earth.

2.2.2. K-Means

The K-means algorithm, mainly known as an unsupervised clustering method, can also be utilized for classification and detection tasks by using its clustering capabilities [40]. In the context of classification, K-means can assign class labels to unlabeled data on the basis of its similarity to predefined centroids. This process involves two steps: training and prediction. During the training phase, a representative set of labeled samples is used to create a set of centroids (cluster centers) that correspond to different classes. Each centroid represents the mean feature vector of its respective class. Features can be extracted using

various techniques, depending on the type of data, such as image descriptors or numerical attributes [41]. In the prediction phase, K-means uses trained centroids to classify new unlabeled samples. The algorithm calculates the Euclidean distance between each unlabeled sample and the centroids, assigning the sample to the class represented by the nearest centroid. The assigned class label can then be used for classification purposes. K-means can also be used for detection tasks by exploiting the notion of outlier detection. In this scenario, the algorithm is trained on a set of normal samples that represent the target class or classes. During prediction, if a new sample is significantly different from the nearest centroid (based on a predefined threshold), it is considered an outlier or anomaly. This approach is useful for detecting abnormal instances or objects that do not conform to the learned patterns. However, it is important to note that K-means have certain limitations when used for classification and detection. One limitation is that it assumes a fixed number of clusters (K) and requires prior knowledge of the number of classes or anomalies in the data. Additionally, K-means are sensitive to the initial placement of centroids and can converge to suboptimal solutions. It also assumes that the clusters are convex and have equal variance, which may not always hold in complex data distributions [42]. For atmospheric studies, it is important to note that the effectiveness of using the K-means algorithm for cloud and aerosol classification and detection depends on the quality and representativeness of the features extracted from the lidar data. Additionally, the choice of the number of clusters (K value) in the algorithm should be carefully considered, as this determines the level of granularity in the classification results. Iterative refinement and validation techniques can be used to optimize clustering and detection performance [43]. K-means has found various applications in the realm of atmospheric science, elucidating patterns and structures in meteorological and climatological datasets that were not immediately apparent. Within the domain of atmospheric science, this method has been used to categorize weather patterns, identify synoptic regimes, and even discern different types of air mass. For example, researchers might use the clustering of K-means in multiyear atmospheric profiles (comprising variables such as temperature, humidity, pressure, and wind speed) to discern predominant atmospheric states or patterns [44]. This categorization can provide invaluable information on typical meteorological conditions associated with specific regions and seasons. Moreover, the K-means algorithm has also been applied to satellite-derived datasets, such as aerosol optical depth or cloud cover, to identify spatial patterns or classify specific clouds or aerosol types based on their spectral properties [45]. The advantage of K-means in these applications is its ability to handle large datasets, often spanning multiple decades, allowing scientists to discern long-term climatological patterns. However, while the technique offers significant utility, its efficacy is based on appropriate preprocessing of data and the judicious selection of k, which remains a subject of ongoing research. As computational capacities grow and datasets become more extensive and intricate, the adoption of K-means clustering in atmospheric science is poised to expand, promising deeper insights into the complex dynamics of the atmosphere. For atmospheric studies and in general, variables with different units can introduce a significant challenge. The Euclidean distance, a fundamental metric in cluster space, may be adversely affected by these disparate units, potentially leading to skewed results. Normalizing the variables to align to a consistent scale diminishes the undue influence of varying units on the Euclidean distances. It is recommended to use feature scaling techniques that standardize variables to possess a mean of zero and unit variance. By adopting this approach, the dominance of variables with larger units in the distance computations is circumvented, thereby promoting an equitable consideration of all variables. This solution not only mitigates the challenges posed by different units but also improves the robustness and accuracy of K-means clustering outcomes.

### 2.2.3. Random Forest

Random forest is a versatile machine learning algorithm that can be effectively used for classification and detection tasks. In the context of classification, random forest constructs an ensemble of decision trees, where each tree is trained on a random subset of input

features and data samples. The algorithm works by iteratively splitting the data based on the selected features, with the aim of maximizing the difference between classes in the resulting subsets. During the classification phase, the group of decision trees collectively votes on the class label for a given input, and the majority class label is assigned as the final prediction [46]. Random forest's ability to combine multiple decision trees and aggregate their predictions makes it robust against overfitting and noisy data. It can handle high-dimensional feature spaces and provides an estimate of feature importance, aiding in understanding the relative contributions of different features to the classification task. Random forest has been successfully applied to various classification problems, such as image recognition, text classification, and medical diagnosis, where it can accurately handle complex decision boundaries and capture non-linear relationships between features and classes. In the context of detection, random forest can be used as a robust algorithm to identify specific objects or patterns within data. For example, in object detection tasks, random forest can be used to detect objects by learning a set of features that describe the appearance or shape of the target object. Different studies suggest that random forest algorithms are used in atmospheric science. For air quality forecasting, RF algorithms are used in conjunction with meteorological models for air quality forecasting [47] or to model atmospheric concentrations of pollutants such as sulfur dioxide [48], and to analyze spatial patterns in forest attributes [49] and flood risk [50].

### 2.2.4. Gradient Boosting Tree

Gradient boosting trees (GBT) can be effectively utilized for classification and detection tasks, due to their ability to handle complex, non-linear relationships in the data and provide accurate predictions. GBTs are an ensemble learning method that combines multiple decision trees to create a robust predictive model. In the context of classification, GBTs can assign labels or classes to input data according to their features. The GBT training process involves sequentially adding decision trees, where each subsequent tree is trained to correct the errors made by the previous trees [51]. During training, the GBT algorithm focuses on samples that were misclassified or had high prediction errors, adjusting the model weights to prioritize those samples [52]. By iteratively improving the model predictions, GBTs can achieve high precision in classifying complex data. GBTs can automatically select relevant features and assign appropriate weights to them during the training process, effectively handling feature interactions that may be missed by linear models. Recent studies suggest that gradient-boosted decision tree algorithms can be used in atmospheric science to improve the performance of atmospheric chemistry transport models [53] and accurately predict meteorological parameters such as air temperature and humidity [54].

### 3. Discussion

*3.1. Aerosol Layers Detection*

In their research article, McGill et al. [55] explored the application of convolutional neural network algorithms for the detection of aerosol and cloud optical and geometric properties features from lidar remote sensing observations. In fact, lidar technology provides vertically resolved profiles of cloud and aerosol features, making it a valuable tool for atmospheric modeling and forecasting. The traditional approach to determining aerosol and cloud geometrical properties involves thresholding techniques, which can be subjective and arbitrary. To improve accuracy and reduce latency, the authors proposed the use of AI/machine learning-based algorithms for real-time detection of atmospheric features. The study focuses on the Cloud Physics Lidar (CPL, [56]) instrument flown on the NASA ER-2 high-altitude aircraft during the IMPACTS field campaign [57]. The CPL data, collected at a high resolution of 10 Hz, were processed in real-time using the CNN algorithm implemented on a Nvidia Jetson TX2 board. The CNN algorithm detected aerosol layers and clouds of the composite image of lidar profiles and output a binary image representing the presence or absence of features. The algorithm's performance was compared to the traditional processing method, and a confusion matrix analysis demonstrated a high level

of agreement between the CNN-derived results and the traditional method. The results indicated that the real-time AI/ML-based approach provided a comparable performance to post-flight processing, even when operating on noisier and lower signal-to-noise ratio (SNR) data. The CNN algorithm enabled true real-time layer detection at native data resolution, significantly improving the availability of data products and increasing horizontal resolution. A confusion matrix was used to evaluate the performance of the AI/ML algorithm in a supervised learning context. This matrix contrasted the AI/ML-generated outcomes with a reference, which in this context was derived from the results produced using the conventional processing method (noting that even this method was not entirely accurate). On 22 February 2022, there was a strong alignment between the two methods, with more than 98% agreement for the "no-layer" classifications and 89% for the classifications containing layers. This confusion matrix is illustrated in Figure 3 of MCGill publication. It is crucial to highlight that the conventional technique processed 50 averaged profiles, which improved the signal-to-noise ratio (SNR), while the AI/ML approach analyzed individual profiles. This represents a limitation of the AI/ML method; processing a single profile can lead to misdetections of layers, due to a lower SNR.

The authors highlighted the potential application of these techniques to spaceborne lidar data, where real-time data and products are crucial for forecasting models. They also discussed future research directions, including classification of detected layers into specific aerosol and cloud types. Overall, this study demonstrated the effectiveness of AI/ML techniques, specifically CNN algorithms, for real-time feature detection and height determination in lidar data. Further advances in real-time layer classification hold promise for enhancing the accuracy and utility of lidar-based atmospheric observations, especially from satellite platforms.

Yorks et al. [58], in 2021, also explored the application of machine learning techniques, specifically CNN, to enhance cloud and aerosol detection in lidar observations obtained from the spaceborne Cloud-Aerosol Transport System (CATS [57]) instrument on the International Space Station (ISS). By employing a wavelet denoising technique during the day, the signal-to-noise ratio (SNR) of CATS lidar data was significantly improved, allowing for the detection of finer horizontal resolution layers. The CNN technique further improved layer detection and cloud-aerosol discrimination, always during daytime operations when the solar background was stronger, enabling better identification of cloud edges and tenuous cloud features. However, the CNN technique exhibited limitations during night, particularly in detecting subvisible features, such as thin optically shaped aerosol layers. Comparisons between the CNN technique and the operational CATS Cloud-Aerosol Detection (CAD) algorithm revealed a good agreement for clear-air bins but showed discrepancies between the cloud and aerosol classifications, especially during the day. Overall, the integration of CNN techniques and wavelet denoising enhanced the CATS lidar data by increasing the SNR, improving layer detection, and enabling the detection of more atmospheric features at finer horizontal resolutions. An example of how the CNN algorithm improved detection during the day compared to the operational CATS algorithm is represented in Figure 2. The study suggested combining traditional techniques with machine learning methods for future lidar data processing, to improve the accuracy of detection of aerosol layer and cloud properties, particularly during the day at finer horizontal resolutions, which is relevant to the air quality community. The findings emphasized the importance of machine learning tools for future lidar missions, enabling the generation of near-real-time (NRT) data products with shorter latencies and improved accuracy.

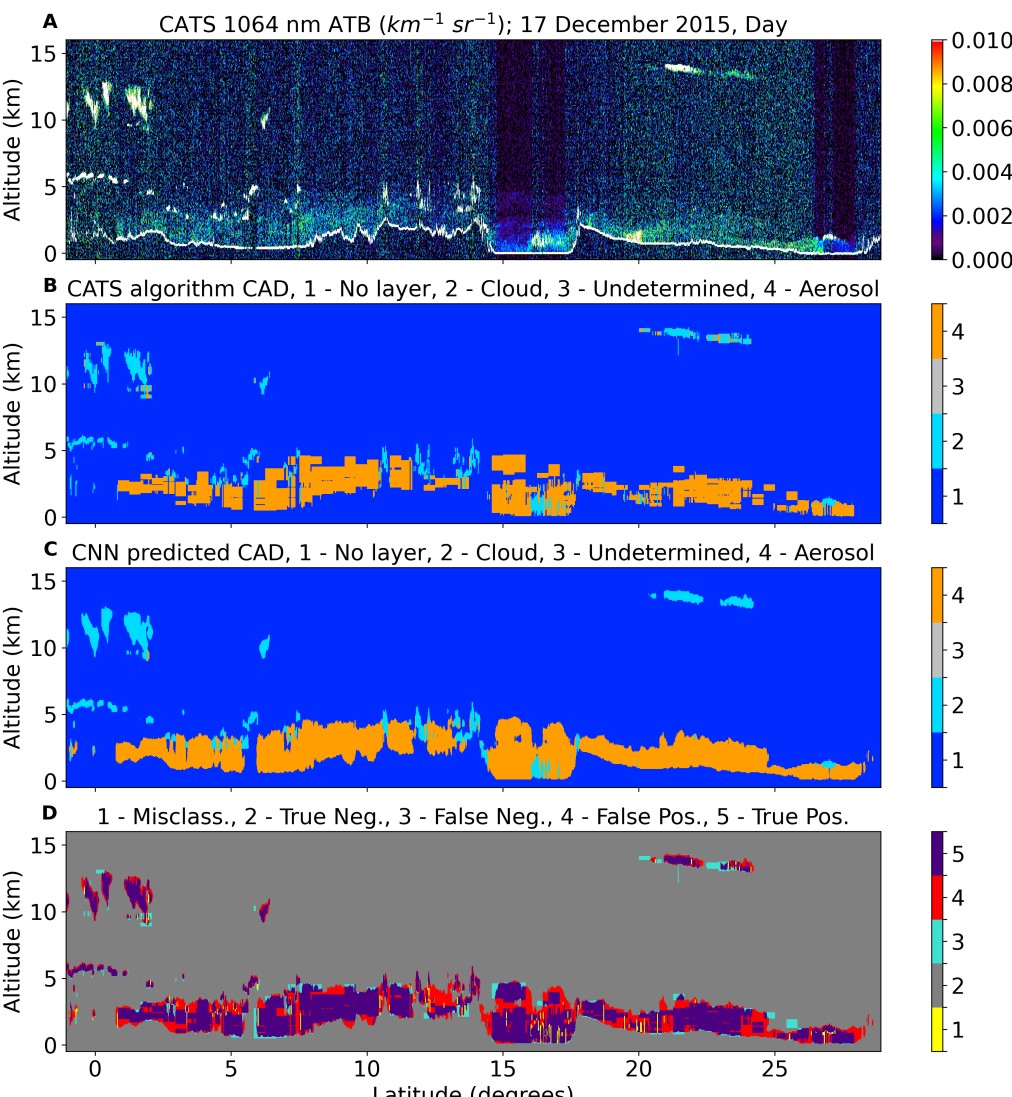

**Figure 2.** Comparison (**A**) CATS Attenuated Backscatter composite image (**B**) CATS cloud–aerosol discrimination (CAD) algorithm Implemented algorithm (**C**) Predicted cloud–aerosol discrimination (CAD) algorithm using CNN (**D**) Differences between (**B**) and (**C**) considering (**B**) as reference between CNN algorithm and operational CATS algorithm in detecting clouds and aerosols. CNN algorithm improved daytime detection.

### 3.2. Aerosol Classification

Zeng et al. [59] presented a study exploring the use of CNN to classify aerosol subtypes in lidar observations collected by the CALIOP instrument (Cloud-Aerosol Lidar with Orthogonal Polarization, [60]) on the CALIPSO (Cloud-Aerosol Lidar and Infrared Pathfinder Satellite Observations, [60,61]) satellite. The goal was to improve the accuracy of aerosol classification by leveraging the texture information inherent in the lidar measurements. The authors compared the performance of their CNN-based classification method with the existing CALIOP operational scene classification algorithms (COSCA). They evaluated the agreement between the CNN classifier and the COSCA in identifying aerosol subtypes, both in the troposphere and in the stratosphere. Using a three-month training set of measurements, the CNN classifier achieved a reasonable level of agreement with COSCA. In the troposphere, the CNN classifier identified the same aerosol subtypes as COSCA in approximately 62% of the cases, while in the stratosphere, the agreement was around 87%. An advantage of the CNN classifier was its ability to identify multiple subtypes of aerosols occurring within the same layer, while COSCA were limited to identifying homogeneous

layers. In the cases where there was a mixture of different types of aerosols, the CNN method provided more accurate results. The study also highlighted some challenges and limitations of the classification process. The broad definitions of aerosol species used by COSCA and the reliance on specific discriminator parameters can introduce biases in classification. Furthermore, the paper discussed the difficulty in distinguishing between certain aerosol subtypes, such as polluted dust and smoke, due to their proximity to geographical locations and the lack of specificity in the class definitions. The performances of the implemented algorithm can be evaluated in [59], Figures 3–6 of Zheng article. The authors suggested future improvements, such as increasing CNN training sets with more accurate field campaign measurements and developing a CNN model of three-dimensional semantic segmentation that includes geographic information. These enhancements aim to further improve the accuracy of aerosol subtype classification, particularly in transition zones between major aerosol source regions, where mixing poses significant challenges. In summary, the paper demonstrated the potential of CNN and deep learning techniques to improve the classification of aerosol subtypes from lidar observations. Using texture information and overcoming the limitations of existing methods, the CNN classifier provides a more realistic representation of aerosol types within individual layers. The findings contributed to the advancement of aerosol characterization and have implications for various applications, including atmospheric modeling and climate studies.

Nicolae et al. [62] described a methodology for classifying aerosols based on lidar data using artificial neural networks (ANN). An aerosol model was developed to compute the optical properties of different types of aerosols. The model consists of six types of pure aerosol: continental, polluted continental, dust, marine, smoke, and volcanic. The model also included different mixtures of these types of aerosol. The aerosol model generated synthetic datasets of lidar measurements by varying the microphysical properties of the aerosols and their composition. Synthetic datasets were used as input for the ANN, which were trained to classify aerosol types based on the lidar data. Various learning rules were tested, including momentum, conjugate gradient descent, step, and Levenberg–Marquardt. The study considered different spectral parameters, such as the extinction Ångström exponent and the backscatter coefficient, to characterize the aerosols. These parameters were calculated on the basis of synthetic datasets. The results showed that the trained ANN achieved a classification accuracy of 75% for more than 80% of the measured data and 75% for more than 90% of the synthetic data. Different ANN architectures were tested, including the multilayer perceptron (MLP), Jordan/Elman networks, generalized feed forward network, self-organizing feature maps, and recurrent neural networks. The study also discussed the uncertainties associated with lidar measurements and synthetic data, emphasizing the importance of considering relative errors in optical parameters. It highlighted the need for a trade-off between the finesse of the output aerosol types and the computing time. In summary, the presented methodology demonstrated the use of artificial neural networks for classifying aerosol types based on lidar data, utilizing a developed aerosol model and synthetic datasets. The results showed promising classification accuracy, offering a potential tool for studying aerosol properties and their impact on the atmosphere.

According to Yang et al. [63], Doppler wind lidars have gained global recognition for their role in wind monitoring and, more recently, aerosol detection. Automatic classification of lidar signals obtained from lidar observations is a challenging task. In this study, the authors explored the capabilities of machine learning in classifying backscattered signals from Doppler wind lidars deployed at the international airport of Reykjavik and Keflavik in Iceland. Their approach involved integrating supervised and unsupervised machine learning algorithms with traditional lidar data processing methods. Through the training of two models, their objective was to effectively filter out noise signals and categorize Doppler lidar observations into distinct classes, including clouds, aerosols, and rain. To initiate the process, they employed machine learning techniques to classify lidar signals. The initial challenge consisted of constituting a dataset that labeled the lidar data to be used as training datasets by the supervised machine learning algorithm. To address this, they combined

the unsupervised data clustering method DBSCAN with a conventional threshold-based approach and manual correction. This combined method was applied to classify four days of data obtained during two dust events in 2019. Subsequently, the labeled dataset was utilized to train two models: a noise discrimination model and a classification model using the RF algorithm. In their study, conducted in Iceland, traditional noise identification in lidar data, based on carrier-to-noise ratio (CNR) thresholds, was found to be suboptimal. In particular, there was not always a linear correlation between the data quality and CNR values, leading to inconsistencies in noise determination. In certain scenarios, CNR values can uncharacteristically spike, rendering typical CNR filtering ineffective. This can result in challenges distinguishing between high clouds and noise, as the cloud measurements' CNR values, especially for higher-altitude clouds, are usually lower than boundary layer signals. To address these issues, the authors introduced a new method of noise identification. After an initial CNR filtering, the DBSCAN clustering method was applied. This approach categorizes data points by their density, effectively distinguishing noise from non-noise, due to their significant density differences. This refined method successfully isolated clouds significantly distant from the boundary layer, labeling them as "high clouds'.' Furthermore, for the purpose of training a supervised machine learning model, particularly a random forest (RF) model, lidar data were compartmentalized into eight groups: low clouds, high clouds, rain, three aerosol types distinguished by their origins and interaction with atmospheric water vapor, other non-noise elements, and noise. This categorization was based on distinct physical parameters, such as CNR values and DBSCAN clustering. For example, high clouds were demarcated by their distance from the boundary layer and isolated using the DBSCAN method, while different types of aerosols were classified based on backscatter and depolarization ratio variances influenced by atmospheric moisture levels. Rain was among the most challenging to label, due to the Doppler lidar's insensitivity to it. Rain classification was based on its low depolarization ratio and descending motion, complicated by instances of high background depolarization ratios. There were cases where it became difficult to distinguish between low clouds and precipitation. Any errors in labels, such as those marked as rain without the associated descending movement, were manually corrected.

The dataset, comprising more than 7.8 million data points, was labeled according to these categories. To train the RF model, four features were selected: height, CNR, backscatter coefficient, and depolarization ratio. After a system breakdown and re-calibration in July 2019, there was a slight variance in the absolute measured values between the datasets from June and July. Therefore, lidar data were normalized daily to maintain consistency. Due to the massive size of the dataset, a 10% random selection was used for model training.

In conclusion, with the trained noise discrimination and classification models in place, the lidar data were processed in two steps. First, CNR was used as input for the noise discrimination model, to segregate noise from non-noise data. Second, height, normalized CNR, backscatter coefficient, and depolarization ratio were used as inputs for the classification model, categorizing the non-noise data points from the first step. Enhancing the accuracy of lidar data labeling would contribute to further improvements in the classification model. Additionally, the performance of the model could be improved by employing a larger and more diverse dataset that encompassed different weather conditions. Additionally, the calibration and correction of the lidar data played a role in the result accuracy; that is, in certain cases, an artifact layer at approximately 600 m was discovered due to the absence of focal correction. On the basis of their findings, they concluded that the continuous incorporation of new training data would further enhance the results. Consequently, the algorithms presented in this study hold promise for real-time interpretation of lidar observations, providing valuable information to end-users such as aviation service providers and air quality observers.

*3.3. Mixing Height Retrieval*

The work of Rieutord et al. [64] focused on developing two machine learning algorithms, K-means for the atmospheric boundary layer (KABL) and AdaBoost for the atmospheric boundary layer (ADABL), to estimate the height of the mixing layer (MLH, [27,65]) from aerosol lidar measurements. The algorithms were applied to a two-year dataset from the Météo-France operational elastic lidar network, and their performance was compared to radiosonde measurements, considered as a reference. The KABL algorithm is a non-supervised algorithm based on the K-means clustering algorithm. It looks for a natural separation in the backscatter signals between the boundary layer and the free atmosphere. The main findings related to KABL were as follows:

- KABL generally outperformed the manufacturer's algorithm, but its performance varied between sites. At the Trappes site, KABL performed well, while at the Brest site, the manufacturer's algorithm performed better;
- The diurnal cycle of the KABL estimates showed a behavior similar to the manufacturer's algorithm but consistently yielded estimates higher by approximately 200 m;
- KABL frequently misidentified the top of the boundary layer, often confusing it with other surface layers or clouds;
- KABL showed versatility by being less dependent on instrumental settings and calibrations, which makes it compatible with backscatter profiles from other instruments.

ADABL is a supervised algorithm based on the AdaBoost algorithm. It fits a large number of decision trees into a labeled dataset and aggregates them intelligently to provide accurate predictions. The main findings related to ADABL were as follows:

- ADABL generally outperformed both KABL and the manufacturer's algorithm at both sites, exhibiting higher correlation and lower error;
- ADABL generated the most pronounced diurnal cycle of the MLH estimates, with a pattern that closely resembled the expected diurnal cycle;
- ADABL's performance was highly dependent on the day it was trained on, with the sunset and sunrise times of those days having an over-influential effect on the estimation;
- ADABL showed promise but had training issues that needed to be addressed, such as the need for an enhanced training set with various meteorological conditions.

The authors concluded that the ADABL algorithm showed a higher potential, with a root mean square error (RMSE) of 550 m with respect to the radiosonde at Trappes vs. 800 m for the manufacturer. On the other hand, the KABL algorithm exhibited a lower performance, with an RMSE of 800 m at Trappes, but it offered greater versatility compared to ADABL. The manufacturer's algorithm, which used a wavelet covariance transform, performed well with minimal tuning, but was not open source. Future developments for ADABL and KABL could include enhancing the training set for ADABL, enforcing time and altitude continuity in KABL estimation, and comparing both algorithms with high-temporal-resolution radiosonde measurements.

Another study by Sleeman et al. [66] addressed the importance of the planetary boundary layer height (PBLH, [25]) for air pollution research and also the limitations of current weather forecast models in accurately predicting PBLH, particularly during night conditions. By applying the CNN algorithm for edge detection and denoising autoencoders for preprocessing, the proposed approach was effective in retrieving PBLH from lidar observations with higher accuracy with respect to classic methodologies. The findings demonstrated that the deep learning method aligned well with radiosonde measurements and outperformed traditional techniques such as wavelet methods. Furthermore, the integration of machine learning-derived PBLH into weather models, specifically the Weather Research and Forecasting with Chemistry (WRF-Chem model), improved the overall model accuracy and provided a breakthrough in air quality forecasting. These findings highlight the potential of the CNN algorithm to improve boundary layer height detection and its application to improve air quality assessments and regional weather forecasting systems.

In 2021, Palm et al. [67] investigated the retrieval of PBLH using satellite lidar data and compared the results with weather model reanalysis data and radiosonde measurements. The study primarily focused on observations obtained from the Ice, Cloud and Land Elevation Satellite (ICESat-2) and CATS instruments, using two retrieval techniques: a classic threshold method and a CNN method. The main findings of the study showed an ability to retrieve the PBLH at a global scale from spaceborne lidars. The threshold method, which sets a backscatter threshold value and analyzes lidar profiles, demonstrated good agreement with radiosonde data, confirming the conventional definition of the PBL top based on temperature inversion and moisture decrease. Furthermore, a comparison with the MERRA-2 reanalysis showed spatial correlations but also highlighted differences, particularly over oceans and specific land areas. The CNN method, based on machine learning algorithms, provided a higher horizontal resolution and showed promising results in retrieving the PBL height. Although some discrepancies were observed compared to the threshold method, it was suggested that the CNN technique may be more accurate in certain cases, as it can work with raw, uncalibrated data. The limitations of satellite lidar data were also stated, including challenges associated with cloud attenuation, daytime noise, and sparse spatial coverage. Despite these limitations, the study emphasized the potential of lidar instruments, especially with improvements to algorithms, in future PBL observing systems. In summary, the threshold method and CNN technique demonstrated their effectiveness, with the former exhibiting good agreement with the radiosonde data and the latter offering a higher horizontal resolution. Although discrepancies were observed compared to the MERRA-2 reanalysis, the general spatial patterns were found to be similar. The limitations of lidar data were recognized, but the study emphasized the potential of lidar instruments to contribute significantly to future PBL observation systems.

### 3.4. Wind Speed and Aerosol Optical Depth

Murphy and Hu [68] introduced an ANN retrieval algorithm to estimate the wind speed of the ocean surface and the optical depth of aerosols using CALIPSO lidar measurements. A neural network algorithm was trained with CALIPSO ocean surface and atmospheric backscatter data, combined with collocated AMSR-E (Advanced Microwave Scanning Radiometer-EOS) ocean surface wind speed data. The study demonstrated the effectiveness of the ANN algorithm by comparing the retrieved wind speeds with AMSR-E and AMSR-2 (Advanced Microwave Scanning Radiometer 2), a passive microwave radiometer onboard the GCOM-W1 (Global Change Observation Mission 1st—Water) satellite, which is operated by JAXA, the Japan Aerospace Exploration Agency. The neural network algorithm was tested by applying it to CALIPSO measurements for the entirety of 2008, to retrieve ocean surface wind speed. The results showed that the retrieval was unbiased, with an average wind speed difference between CALIPSO and AMSR-E of around 0.001 m/s. The standard deviation of the wind speed difference between CALIPSO and AMSR-E, which was considered an instantaneous error, was around 1.05 m/s. The retrieved CALIPSO wind speeds agreed with AMSR-E for both low and high wind speeds. The neural network also automatically adjusted aerosol lidar ratios, minimizing wind speed retrieval errors using ocean surface and atmospheric signals. This eliminated the need to assume predetermined lidar ratios, making the algorithm more versatile with respect to the traditional inversion methods. The authors used the MATLAB embedded functional fitting neural network function, fitnet. The input variables for the neural network included various CALIPSO measurements related to the ocean surface and atmospheric backscatter, as well as latitude. The output variable was the wind speed of the ocean surface. The study highlighted the importance of turbulence in the exchange of momentum, energy, and greenhouse gases between the atmosphere and the ocean. The retrieval of high spatial resolution ocean surface wind speed from CALIPSO lidar measurements contributed to understanding the vertical turbulence exchange and air–sea interaction. Additionally, the algorithm allows for the derivation of aerosol optical depths (AOD) using CALIPSO's ocean surface backscatter signal and theoretical ocean surface reflectance calculated from wind speed

and wind–surface slope variance relation. The study emphasized the applicability of the algorithm to most clear skies with optical depths up to 1.5, without assuming aerosol lidar ratios. The findings contributed to advances in aerosol distribution research, climate dynamics, and the accuracy of wind speed and aerosol optical depth retrievals.

## 4. General Considerations, Open Issues, and Future Perspectives

The machine learning techniques applied for detecting aerosols and clouds from lidar observations are still in their early stages but hold great potential for future advancement. In Table 3, all the different machine learning algorithms implemented to detect, classify, and study atmospheric aerosol layers, clouds, and boundary layers are summarized, together with the observation platform and lidar type. This table highlights that the detection and classification of aerosols has received much more attention with respect to clouds. ML algorithms were developed mostly for satellites and aircraft observations. Algorithm applications to networks of lidars have mostly been dedicated to mixing and planetary boundary layer detection.

**Table 3.** D = Detection; C = Classification; CNN = Convolution Neural Network; RF = Random Forest; GBT = Gradient Booster Trees; ANN = Artificial Neural Network; KM = K-Means; HSRL = High Spectral Resolution Lidar [20]; DWL = Doppler Wind Lidar [21]; EL = Elastic Lidar [5]; RL = Raman Lidar [12]; MLH = Mixing Layer Heigh; PBLH = Planetary Boundary Layer Heigh; AOD = Aerosol Optical Depth.

|  | Type and Refs | Platform | Lidar | Network |
|---|---|---|---|---|
| Aerosols | D, C - CNN; McGill [55] | Aircraft | HSRL | No |
|  | D - CNN; Yorks et al. [58] | Satellite | EL | No |
|  | C - CNN; Zeng et al. [59] | Satellite | EL | No |
|  | D - RF, GBT; Yang et al. [63] | Ground | DWL | No |
|  | C - ANN; Nicolae et al. [62] | Ground | ML | Yes |
| Clouds, Rain | C - RF, GBT Yang et al. [63] | Ground | DWL | No |
|  | D - CNN; Yorks et al. [58] | Satellite | HSRL | No |
| PBLH, MLH, AOD | AOD - ANN Murphy, Hu [68] | Satellite | EL | No |
|  | MLH - KM, GBT Rieutord et al. [64] | Ground | EL | Yes |
|  | PBLH - CNN, Sleeman et al. [66] | Ground | EL | Yes |
|  | PBLH - CNN, Palm et al. [67] | Satellite | EL | No |

As a general remark for future algorithm implementation, it is important to exercise caution when applying existing machine learning algorithms without proper consideration. It is crucial to always consider the underlying physics processes that govern the interaction of light with atmospheric constituents. Machine learning models must always be designed and trained not as a black box, but based on correct reasoning, taking into account real-world relationships, to ensure accurate and meaningful results.. This ensures that trained models are capable of distinguishing between various aerosol species, such as dust, smoke, and pollution, as well as different types of clouds, including liquid and ice clouds. Additionally, understanding the physics behind lidar measurements enables the development of machine learning algorithms that can exploit different lidar techniques, which will extract relevant information from lidar profiles, such as backscatter, extinction, depolarization ratio, and spectral characteristics, to enhance the accuracy of classification. Although machine learning techniques offer the potential to automate aerosol and cloud detection, it is important to recognize the limitations and uncertainties associated with these approaches. Indeed, machine learning algorithms should be trained on diverse and representative datasets that cover a wide range of atmospheric conditions and aerosol/cloud types, to enhance their robustness and applicability. Furthermore, regular validation and evaluation against independent measurements and ground-based observations are necessary, to assess the accuracy and reliability of the classification results.

## 5. Conclusions

Machine learning applied to lidar observation shows promising results in the detection and classification of aerosols and clouds. Advanced algorithms such as convolutional neural networks (CNN), artificial neural networks (ANN), random forests (RF), and gradient boosting trees (GBT) have shown significant improvements in accuracy and efficiency. These techniques enable real-time and automated analysis of lidar data, providing valuable insights into atmospheric phenomena. However, there is still a need for future research in this field, particularly to test the performance of machine learning algorithms over multiple instruments, i.e., a lidar network and/or a research infrastructure, for an extended period of time. Continuous monitoring of aerosols and clouds is crucial for understanding their dynamics, impact on climate, and air quality. Therefore, evaluating the performance and reliability of machine learning algorithms across multiple lidar systems over long durations is essential to ensure their robustness and generalizability. Moreover, future research should focus on expanding the capabilities of machine learning algorithms in lidar data analysis. This includes exploring new techniques, such as deep learning architectures, to further improve the accuracy and efficiency of detection and classification tasks. Furthermore, incorporation of additional data sources, such as meteorological data and satellite observations, could improve the contextual understanding of aerosols and clouds, leading to more comprehensive and accurate analyses. By addressing these research needs, we can advance the field of machine learning applied to lidar observation for the detection and classification of aerosols and clouds. This would contribute to a better understanding of atmospheric processes, climate change, and air quality, which would ultimately lead to better environmental monitoring and decision making.

**Funding:** This research received no external funding.

**Data Availability Statement:** Not applicable.

**Conflicts of Interest:** The author declares no conflict of interest.

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
