# Peer review of "Machine Learning Techniques for Vertical Lidar-Based Detection, Characterization, and Classification of Aerosols and Clouds: A Comprehensive Survey"

_remotesensing, doi:10.3390/rs15174318_

Round 1

Reviewer 1 Report

 “Machine Learning Techniques for Lidar-Based Detection, Characterization, and Classification of Aerosols and Clouds: A Comprehensive Survey” reviewed recent work on applying Machine Learning techniques on lidar sensors, including aerosol/cloud detections, classification, and retrievals, and planetary boundary layer (PBL) height detection.  Recent couple years there have been many attempts of using ML on atmospheric science and a good survey paper that can summarize the existing efforts are welcomed so that researchers can continue explore the possibility of ML on lidar applications while avoiding the lessons learnt from previous studies. The article, however, has several places that are not well explained so that general audience can understand the problem easily.  These issues are listed below:

1.     Line 30 to 31, “high spatial and temporal resolution observations”. This is confusing as lidar’s benefit is vertical resolving observations instead of high spatial and temporal resolution. As lidar had narrow swath and thus, have poor revisit time for selected phenomenon. 

2.     Line 53 to 54, The separation of first and second class are not well defined. For one to classify aerosol vs cloud, it needs to detect feature boundary first. Thus, the two class are somewhat intermingled and can be categorized into one. 

3.     Figure 2 is very confusing as ML is primary another way of extracting information from raw observations and it cannot provide additional information beyond the given input. Thus, it is hard to explain how ML can reduce the gap between Raman lidar and Elastic lidar (EL).  What is more realistic is that ML also make its own assumptions, maybe not as physical as lidar ratio, but something that also contains similar information as lidar ratio into the ML generated relations. The figure also makes it seems like ML is only used for that purpose while later in this paper, ML are applied to both EL and HSRL and Doppler wind lidar (DWL). 

4.     Section 2.2. This section introduced CNN, K-Means, ANN, and other ML techniques. The structure is confusing because CNN is a form of ANN, but they two are listed separately as if they two are two independent methods. Second, some of the ML techniques are described with examples of how they are used in atmospheric science field, while others are only simply descriptions of method itself.  It is expected to be consistent. 

5.     In section 2.1, the DWL is not mentioned, but used in later studies.  It is also expected to list all the existing major sensors, their physics-based algorithm and the algorithm’s advantage and limitations, including the traditional denoising method which are mentioned several times in later chapters. 

6.     Line 180, satellite imagery is part of remote sensing data.

7.     Line 205, all ML is highly dependent on quality and representativeness of the training data, not just ANN.

8.     In section 3 while describing the previous study, it is not listed how do each study generate the ML model, which is crucial for researcher to replicate or improve from the existing study. 

9.     Section 3.1, the accuracy and limitation of ML detecting aerosol layer is not mentioned. 

10.  Line 307-308, “It is worth noting that the definitions of the CALIPSO aerosol class are not discrete, but represent mixture continuums, making the agreement level satisfactory.” This sentence is out of content and should be explained in section 2.1.

11.  Line 309-310, there is no figure show the advantage of CNN detecting multiple subtypes within an aerosol layer. Because this is the biggest advantage of CNN in this study, an illustration will be very helpful. 

12.  Line 368, again how are the noise discrimination model and classification model structured?

13.  Line 399, “KABL tended to oscillate between several candidates for the top of the boundary layer, including surface layers or clouds.” This sentence is very hard to understand.

14.  Line 449, how are “promising results” defined?

15.   Line 454-456, due to lack of PBL background described in section 2, one would think that using vertical resolving lidar to retrieve PBL height is a very logical and common technique. 

16.  Line 469, what is AMSR-2? Also later, what are the benchmark to show that NN results are better than AMSR-E traditional algorithm retrievals?

17.  Line 499-505. To me “taking into account these physics-based relationships” in ML means explainable ML, which is to ensure ML results are correct based on the correct reason. I cannot build relations between that and that model can distinguishing between various aerosol species (to me, this is based on building a representative training sample), and development of ML algorithms on different lidar techniques. 

Author Response

I deeply appreciate the reviewer's insightful feedback and meticulous review. As evident from my expertise in lidar, my primary knowledge base isn't in machine learning. Your guidance has been instrumental in enhancing the quality and clarity of the manuscript in the ML context. Thank you for your invaluable contribution. I took into consideration all your points/comments in drafting the new manuscript. Below point-by-point responses. 

Line 30 to 31, “high spatial and temporal resolution observations”. This is confusing as lidar’s benefit is vertical resolving observations instead of high spatial and temporal resolution. As lidar had narrow swath and thus, have poor revisit time for selected phenomenon. 

        I modified the sentence accordingly to make it more clear

Line 53 to 54, The separation of first and second class are not well defined. For one to classify aerosol vs cloud, it needs to detect feature boundary first. Thus, the two class are somewhat intermingled and can be categorized into one. 

I agree and I merged the two classes into one

Figure 2 is very confusing as ML is primary another way of extracting information from raw observations and it cannot provide additional information beyond the given input. Thus, it is hard to explain how ML can reduce the gap between Raman lidar and Elastic lidar (EL).  What is more realistic is that ML also make its own assumptions, maybe not as physical as lidar ratio, but something that also contains similar information as lidar ratio into the ML generated relations. The figure also makes it seems like ML is only used for that purpose while later in this paper, ML are applied to both EL and HSRL and Doppler wind lidar (DWL). 

     I agree that the figure is not conveying any useful information and  it is also       wrong. I deleted it

Section 2.2. This section introduced CNN, K-Means, ANN, and other ML techniques. The structure is confusing because CNN is a form of ANN, but they two are listed separately as if they two are two independent methods. Second, some of the ML techniques are described with examples of how they are used in atmospheric science field, while others are only simply descriptions of method itself.  It is expected to be consistent. 

   Agreed, I added to each algorithm also the relative applications on          atmospheric science field 

In section 2.1, the DWL is not mentioned, but used in later studies.  It is also expected to list all the existing major sensors, their physics-based algorithm and the algorithm’s advantage and limitations, including the traditional denoising method which are mentioned several times in later chapters. 

Agreed, I added DWL, and also integrated a table where all the techniques are listed, described and explained. 

Line 180, satellite imagery is part of remote sensing data.

Changed accordingly

Line 205, all ML is highly dependent on quality and representativeness of the training data, not just ANN.

Changed accordingly

In section 3 while describing the previous study, it is not listed how do each study generate the ML model, which is crucial for researcher to replicate or improve from the existing study. 

I changed it accordingly adding how the ML models are generated

Section 3.1, the accuracy and limitation of ML detecting aerosol layer is not mentioned. 

It is now added 

Line 307-308, “It is worth noting that the definitions of the CALIPSO aerosol class are not discrete, but represent mixture continuums, making the agreement level satisfactory.” This sentence is out of content and should be explained in section 2.1.

Agreed and moved to section 2.1 

Line 309-310, there is no figure show the advantage of CNN detecting multiple subtypes within an aerosol layer. Because this is the biggest advantage of CNN in this study, an illustration will be very helpful. 

I contacted the authors to ask for a similar image without copyright, but I got no answer, so I added a reference to a figure of their paper. 

Line 368, again how are the noise discrimination model and classification model structured?

A new paragraph now describes how both models are structured

Line 399, “KABL tended to oscillate between several candidates for the top of the boundary layer, including surface layers or clouds.” This sentence is very hard to understand.

I rephrased to make it more clear

Line 449, how are “promising results” defined

 Changed accordingly 

Line 454-456, due to lack of PBL background described in section 2, one would think that using vertical resolving lidar to retrieve PBL height is a very logical and common technique. 

Now a full description is added in section 2.1 

Line 469, what is AMSR-2? Also later, what are the benchmark to show that NN results are better than AMSR-E traditional algorithm retrievals?

Agreed that the information was lacking. Now a new paragraph is added making it more clear

Line 499-505. To me “taking into account these physics-based relationships” in ML means explainable ML, which is to ensure ML results are correct based on the correct reason. I cannot build relations between that and that model can distinguishing between various aerosol species (to me, this is based on building a representative training sample), and development of ML algorithms on different lidar techniques. 

Agreed! Now the sentence is rephrased to make it more clear and correct

Reviewer 2 Report

Journal Remote Sensing paper ID 2515970

Review paper: Title

Machine Learning Techniques for Lidar-Based Detection, Characterization, and Classification of Aerosols and Clouds: A Comprehensive Survey

Simone Lolli

The paper proposes reports of machine learning applications to analyse vertical lidar measurements. These techniques enable the detection, classification, and characterisation of aerosols and clouds by leveraging the rich features contained in the lidar signals.

In this article, an overview of different machine learning architectures and algorithms employed in the field isprovided, highlighting their strengths, limitations, and potential applications. Additionally, the survey examines the impact of machine learning techniques on improving the accuracy, efficiency, and robustness of aerosol and cloud real-time detection from lidar observations.

The paper is well written and gives a broad overview of lidar applications in which machine learning techniques are applied for the analysis and characterization of atmospheric information. For this reason, the paper is not an article but should be classified as a short review.

Before accepting the paper for submission it is necessary to make some small changes and improvements.

1.

I would like to underline that just to speak of classification and categorization is needed to say that there are two different lidar application:

-       Vertical Lidar for atmospheric study 

-       Horizontal lidar per environmental and safety and security study

For this the title is not appropriate, and to better describe the content of the paper I suggest:

“Machine Learning Techniques for vertical Lidar-Based Detection, Characterization, and Classification of Aerosols and Clouds: A Comprehensive Survey”

1.     it is necessary to have the paper reread by a native speaker moreover some errors in the text are present.

Only after these last two suggestions satisfy could the paper be accepted for its publication.

Author Response

Thanks for your positive comments. I changed the title accordingly and the manuscript was revised to check English 

Reviewer 3 Report

The attached file contains the comments/suggestions.

Well written with a few errors.

Author Response

We really appreciate the reviewer's work in reading and comment the submitted manuscript. Thanks to her/his comments and suggestions, we are able to provide a largely improved version. Below, answers to point-by-point comments.

Agreed, the article was not going in much deeper analysis. Now text was added to explain more in detail and deeper all the different technologies, paying attention to the methodologies of the different implemented algorithms. Consider that now it is almost 19 pages with a figure removed.

1) It was changed accordingly. All the discussion now follows a different approaches and incorrect claims were removed, to state more correctly what the algorithms do. 

2) Citations were added, even if the text is now different

3) A new paragraph is added to address the issue raised by the meaningful comment of the reviewer.

4) 

a) Thanks for the comment, now it is explained that the ML main feature it is that can work on profile-by-profile basis. This is well suited for a product in near-real time. The traditional method, taken as reference, it works on 50 profiles. ML algorithm achieves almost the same performances using one profile (less than 10% difference). It worths noting that even the traditional method cannot be considered 100% accurate. 

b) The approach proposed by Yang et al., is now explained in much more detail. 

c) Thanks!

d) NN is changed accordingly to ANN all over

5) Thanks, the section also has been expanded to make it more clear. 

Round 2

Reviewer 1 Report

The modified manuscript addressed all previous mentioned issue. One minor thing: line 555, there are two periods at the same location.